# Probing bacterial cell wall growth by tracing wall-anchored protein complexes

Yi-Jen Sun [1,4], Fan Bai [2,4], An-Chi Luo[1], Xiang-Yu Zhuang[1], Tsai-Shun Lin[1], Yu-Cheng Sung[1], Yu-Ling Shih [3] & Chien-Jung Lo [1✉]

The dynamic assembly of the cell wall is key to the maintenance of cell shape during bacterial growth. Here, we present a method for the analysis of *Escherichia coli* cell wall growth at high spatial and temporal resolution, which is achieved by tracing the movement of fluorescently labeled cell wall-anchored flagellar motors. Using this method, we clearly identify the active and inert zones of cell wall growth during bacterial elongation. Within the active zone, the insertion of newly synthesized peptidoglycan occurs homogeneously in the axial direction without twisting of the cell body. Based on the measured parameters, we formulate a Bernoulli shift map model to predict the partitioning of cell wall-anchored proteins following cell division.

[1] Department of Physics and Graduate Institute of Biophysics, National Central University, Jhongli, Taiwan, ROC. [2] Biomedical Pioneering Innovation Center (BIOPIC), School of Life Sciences, Peking University, Beijing, China. [3] Institute of Biological Chemistry, Academia Sinica, Taipei, Taiwan, ROC. [4]These authors contributed equally: Yi-Jen Sun and Fan Bai ✉email: cjlo@phy.ncu.edu.tw

Cell growth and division are essential for life. The maintenance of a stable cell shape during growth and division is vital, and is a tightly controlled process across different biological species[1–4]. In bacteria, the cell envelope comprises 2D-liquid lipid membranes and a rigid peptidoglycan cell wall, the latter of which is a mesh-like structure formed by the peptidoglycan strands crosslinked between short peptide stems[5–7]. In addition to maintaining cell shape, the cell envelope of bacteria is also crucial for providing mechanical strength to balance osmotic pressure[8]. The growth, remodeling, and division of the peptidoglycan layer are dynamic and robust processes assisted by cytoskeletal proteins and enzymes possessing different functions[5,9–13]. While the chemical compositions and structures of peptidoglycans are well studied, the spatiotemporal regulation of peptidoglycan insertion and remodeling during bacterial growth remains poorly understood[5–7,14]. Previous studies have taken advantage of the fact that bacteria incorporate fluorescent D-amino acids into the cell wall during synthesis, demonstrating qualitative measurement of cell wall insertion patterns[7,15,16]. However, the low positional specificity of fluorescent D-amino acids limits further quantitative understanding of cell wall synthesis at high spatial and temporal resolution in live cells.

Herein, we provide a strategy to probe the dynamics of bacterial cell wall growth by tracing the movement of the peptidoglycan-anchored protein complexes, bacterial flagellar motors (BFMs)[17,18], during cell growth and division in *Escherichia coli* (*E. coli*). We identified the active and inert zones of cell wall growth during bacterial elongation and find expansion of the cylindrical cell body is homogeneous in the axial direction. By tracking the BFMs near the cell center, we demonstrate the new cell poles are made by newly synthesized peptidoglycans. We further show the partitioning of cell wall-anchored protein complexes after cell division can be modeled by a Bernoulli shift map, the biological function of which is to smooth uneven protein distributions.

## Results

**Labeling BFMs as fluorescent "landmarks"**. BFMs consist of a bi-directional rotatory motor embedded in the cell envelope, a short proximal hook, and a long extracellular helical flagellar filament[19–21]. We bound streptavidin conjugated Alexa Fluor 594 to the biotinylated hooks of BFMs in *E. coli* (Fig. 1A)[22] (See Methods and Supplementary information), and an average of 8.5 fluorescent spots could be observed per cell (Fig. 1B). Previous structural and biophysical studies have indicated that the trans-envelope BFMs are firmly anchored to the cell wall[17,18]. This was confirmed by our observation that the location of BFMs were static in non-growing cells (Supplementary Fig. 1). We further monitored the rotation of the cell body in a tethered cell experiment when the cell was slowly growing in growth medium. We found that both the rotation speed and geometry of the cell body driven by a BFM were very stable during the cell growth (Supplementary Fig. 2A, B). In addition, in a tethered cell experiment conducted in growth medium, we applied a transient hydrodynamic flow to push the cell body away from the original rational geometry. When the flow was stopped, we found that the rotation speed and geometry of the cell body was almost identical to the position before the flow was applied (Supplementary Fig. 2C). These evidence altogether strongly support that BFMs are firmly anchored to the cell wall during the cell growth. Therefore, we could probe the dynamics of bacterial cell wall growth by measuring the changes in distance between any pair of these fluorescent "landmarks" or between a fluorescent BFM and a cell pole using time-lapse microscopy (Fig. 1B). Our method provides an alternative and valuable approach, since the natural process of bacterial cell wall assembly is not perturbed.

**Inert zone determination**. The rod-shaped *E. coli* can be viewed as a cylinder with a spherical cap at each cell pole. During cell growth, the two caps remain intact and the cylinder elongates. Presumably, the spherical caps are the inert zones and the cylindrical body is the active zone for peptidoglycan insertion;[7,14] however, whether a boundary exists, separating different functional zones, remains unknown. In the present study, we defined the axial position ($P_y$) of each BFM as the distance from its nearest cell pole, and the change in $P_y$ was monitored over time during cell growth (Fig. 1B). Our results showed that the $P_y$ of BFMs nearest the cell pole remained constant, indicating the absence of cell wall synthesis in this zone. In contrast, the $P_y$ of BFMs further away from the cell pole increased as the cell elongated, verifying that they were located in the active zone for cell wall synthesis. For further quantitation, we obtained the average axial velocity ($V_{P_y}$) of each BFM via linear fitting to $P_y$ vs. time traces (Fig. 1C) and plotted $V_{P_y}$ with respect to the initial axial position ($P_{y0}$) of each BFM (Fig. 1D). The data points were subjected to piecewise linear fitting composed of a plateau corresponding to $V_{P_y} = 0$ and a slope with a positive gradient, see Methods (the instantaneous velocity vs. instantaneous axial position was provided in Supplementary Fig. 3, which showed the result consistent with Fig. 1D). The intercept was found at the axial position $P_{y0(LB)} = 0.27 \pm 0.04$ μm for cells grown in LB, therefore, the boundary between the inert and active zones for cell wall growth in live *E. coli* has been determined. It is important to note that the size of the inert zone is smaller than half of the cell diameter. We also found that the size of the inert zone is self-adjusted and correlated to the nutrient quality in the growth medium and cell diameter. The size of the inert zone ($P_{y0(TB)} = 0.22 \pm 0.04$ μm) and the cell diameter ($D_{(TB)} = 1.02 \pm 0.06$ μm) were smaller for cells grown in TB, a less rich growth medium. In contrast, when cells were grown in the rich SOC medium, the size of the inert zone ($P_{y0(SOC)} = 0.37 \pm 0.04$ μm) and the cell diameter ($D_{(SOC)} = 1.11 \pm 0.06$ μm) were increased.

**Homogeneous cell wall insertion in the active zone**. Subsequently, we probed the cell wall growth dynamics within the active zone by monitoring changes in the relative distance (lateral: $D_{xij}$; axial: $D_{yij}$) between paired BFMs (*i*th and *j*th, BFM) imaged on the same cell (Fig. 2A and B). Excluding BFMs located in the inert zone, we calculated the relative velocity (lateral: $V_{Dx} = \Delta D_x/\Delta t = (D_x'-D_x)/\Delta t$; axial: $V_{Dy} = \Delta D_y/\Delta t = (D_y'-D_y)/\Delta t$, $\Delta t = 10$ min) between paired BFMs and plotted $V_{Dx}$ vs. $D_x$ and $V_{Dy}$ vs. $D_y$, respectively. Interestingly, we found that $V_{Dy}$ increased linearly with increases in $D_y$, indicating a homogeneous cell wall growth pattern in the axial direction (Fig. 2C), consistent with previous observations using D-amino acid[7,23]. A simple equation can be used to describe the cell wall growth: $V_{Dy} = H \, D_y$, where $H$ is the 'Hubble's parameter" for bacterial cell wall expansion and indicates the instantaneous growth rate per unit axial distance; $H$ increases as the cell elongates (Fig. 2C, D). The homogenous cell rod expansion can be further confirmed by calculating the cell rod growth velocity and cell rod length from phase-contrast images, (Fig. 2C, Gray dots). The cell rod growth data were on the extrapolation of the fitting curves of $V_{Dy}$ vs. $D_y$ as additional evidence supporting the homogenous expansion of the cell rod. While the axial insertion rate increases as the cell elongates (Fig. 2D), if we normalize each $V_{Dy}$ by $H$, all data fall on the same line indicating uniform cell wall growth at all cells and stages (Supplementary Fig. S4A).

In the lateral direction, however, we did not observe any change in $D_x$ between any pair of BFMs (Fig. 2E, F, and Supplementary Fig. 4B), indicating cell wall growth only occurs in the axial direction in *E. coli* without increasing the diameter or

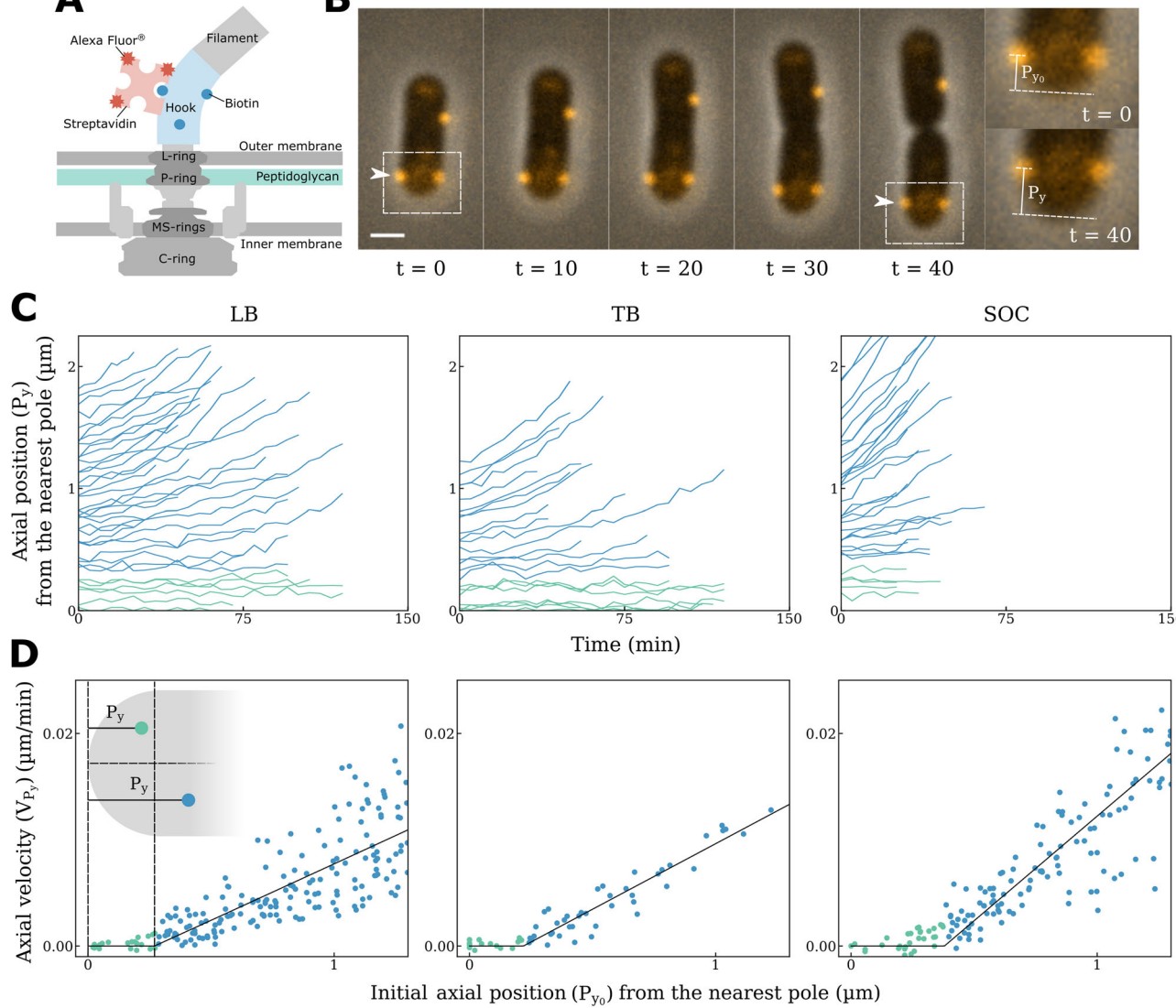

**Fig. 1 Real-time monitoring of *E. coli* cell wall growth using fluorescently labeled flagellar motors (BFMs) as landmarks. A** A schematic showing the fluorescent labeling of a bacterial flagellar hook with Alexa Fluor 594. **B** Time-lapse fluorescence images showing the dynamic movement of BFMs in actively growing cells. Fluorescence and phase-contrast images were overlaid (scale bar, 1 μm; Δt = 10 min). At least 3 times of the same experiment was repeated with similar results. **C** Sample traces tracking the axial positions ($P_y$) of BFMs during cell elongation in different growth media (LB: $n = 45$; TB: $n = 22$; SOC: $n = 31$). Green and blue lines represent BFMs within the inert zone and the active zone, respectively. **D** Average velocity of BFM movement during cell growth identified an inert zone near the cell poles in different growth media. Green dots: BFMs in the inert zone; blue dots: BFMs in the active zone; Total number of tracked flagellar motors, LB: $n = 195$; TB: $n = 56$; SOC: $n = 220$); inset: a schematic showing the $P_y$ of a BFM from the nearest cell pole.

twisting of the cell body, an observation that differs from the previous reports[24]. Our results strongly support the disperse peptidoglycan insertion mechanism underlying the cell wall growth[4,9,14] and impose constrains on the models[25].

**Newly synthesized peptidoglycans form the new poles.** As a result of the observed homogenous axial insertion pattern during cell elongation, we were able to introduce a normalized coordinate to the cylindrical cell body (excluding the two inert zones at the cell poles) by repositioning the origin to the center of the cell (Fig. 3A). The location of each BFM can now be described using the normalized coordinates: axial position, $N_y = Y/L$; where $Y$ and $L$ are the $y$-position of the BFM and length of the active zone, respectively. We plotted $N_y$ with respect to time during cell elongation (Fig. 3B), and all BFMs stayed in the same normalized axial positions (Fig. 3B and C).

As the cell entered the division stage (when a convexity defect began to emerge at the center of the cell), the insertion of newly synthesized peptidoglycan was no longer homogeneous in the axial direction within the active zone due to the formation and constrictive nature of the division septum[7,14]. This was evidenced by the observation that BFMs were driven away from the origin in the normalized coordinate (Fig. 3A). According to our measurements, the $|N_y|$ of BFMs increased over time after the cell entered the division stage (Fig. 3D, E), indicating that extra peptidoglycan insertion occurred locally at the center of the cell due to septum growth. To further investigate this phenomenon, we tracked the movement of BFMs that were near the center of the cell in the initial image frames. We found that this subset of BFMs migrated away from the center of the cell during cell division by up to 0.27 μm (Fig. 3F). The same distance was classified as the inert growth zone (Fig. 1). An interesting case is documented in Fig. 3G, H, in

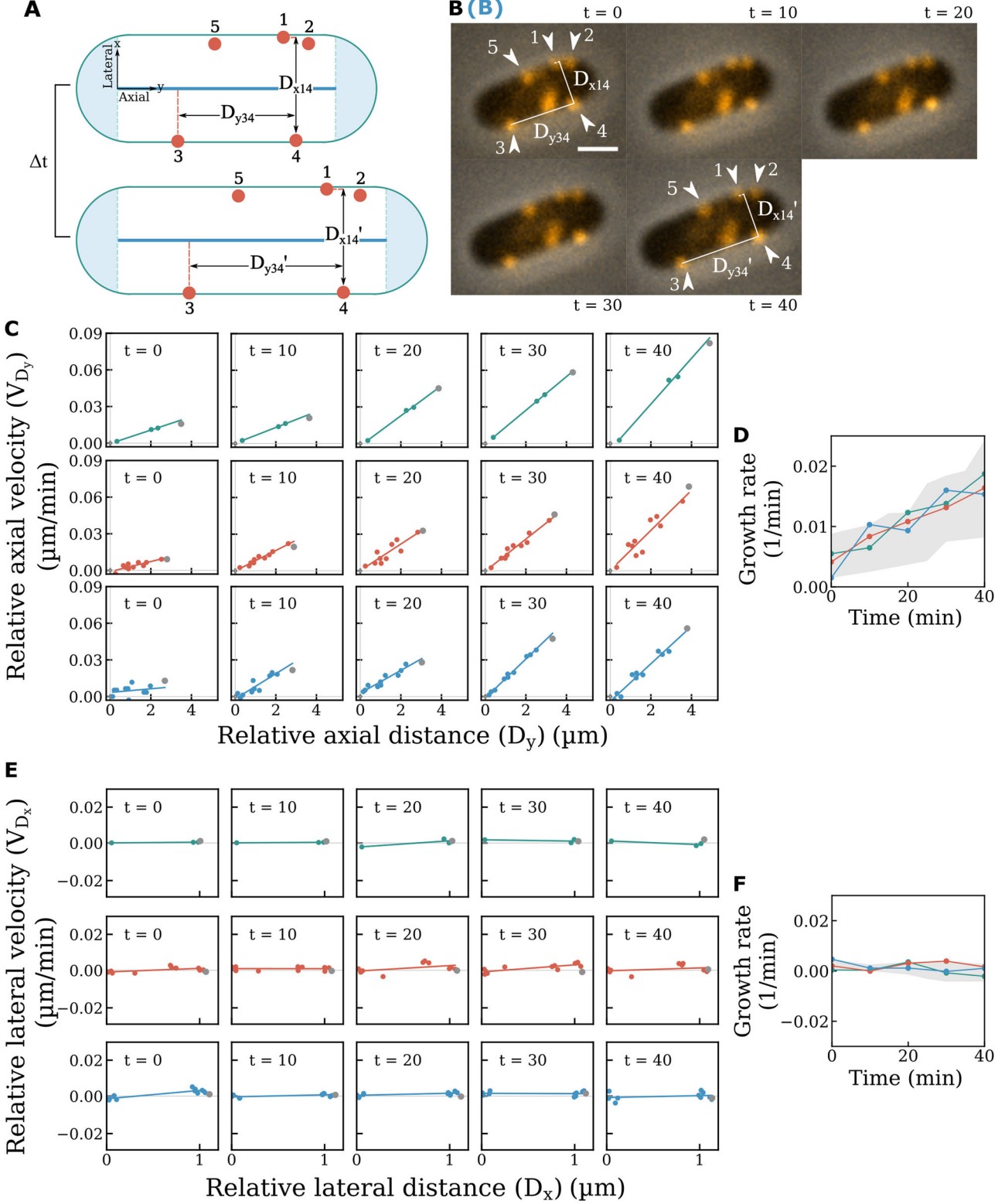

**Fig. 2 Expansion dynamics of the cell wall within the active growth zone. A** A schematic showing changes in the relative distances between BFM pairs during cell elongation. **B** Time-lapse fluorescence images showing changes in the relative distances between BFM pairs in a live cell. Fluorescence and phase-contrast images were overlaid (scale bar, 1 μm; $\Delta t = 10$ min). At least 3 times of the same experiment was repeated with similar results. **C** The $V_{Dy}$ vs. $D_y$ relationship between paired BFMs in three representative cells. Each row shows $V_{Dy}$ vs. $D_y$ between paired BFMs from the same cell at different time points. $V_{Dy}$ vs. $D_y$ displays a good linear relationship at all times, but the slopes are different, $\Delta t = 10$ min. Gray dots represented total cell rod growth velocity vs. cell rod length, which also supported uniform growth of the cell rod. **D** The axial growth rate increases with time, $n = 11$ (number of cells). The colors represent data from different cells as in (**C**). The gray area represents the range of the data. **E** The $V_{Dx}$ vs. $D_x$ relationship between paired BFMs in three representative cells [same cells as in (**C**)]. Each row shows $V_{Dx}$ vs. $D_x$ between paired BFMs from the same cell at different time points. $V_{Dx}$ is ~0 regardless of $D_x$. Gray dots represented total cell width growth velocity vs. cell width. **F** The lateral growth rate was also zero over time, $n = 7$ (number of cells). The gray area represents the range of the data.

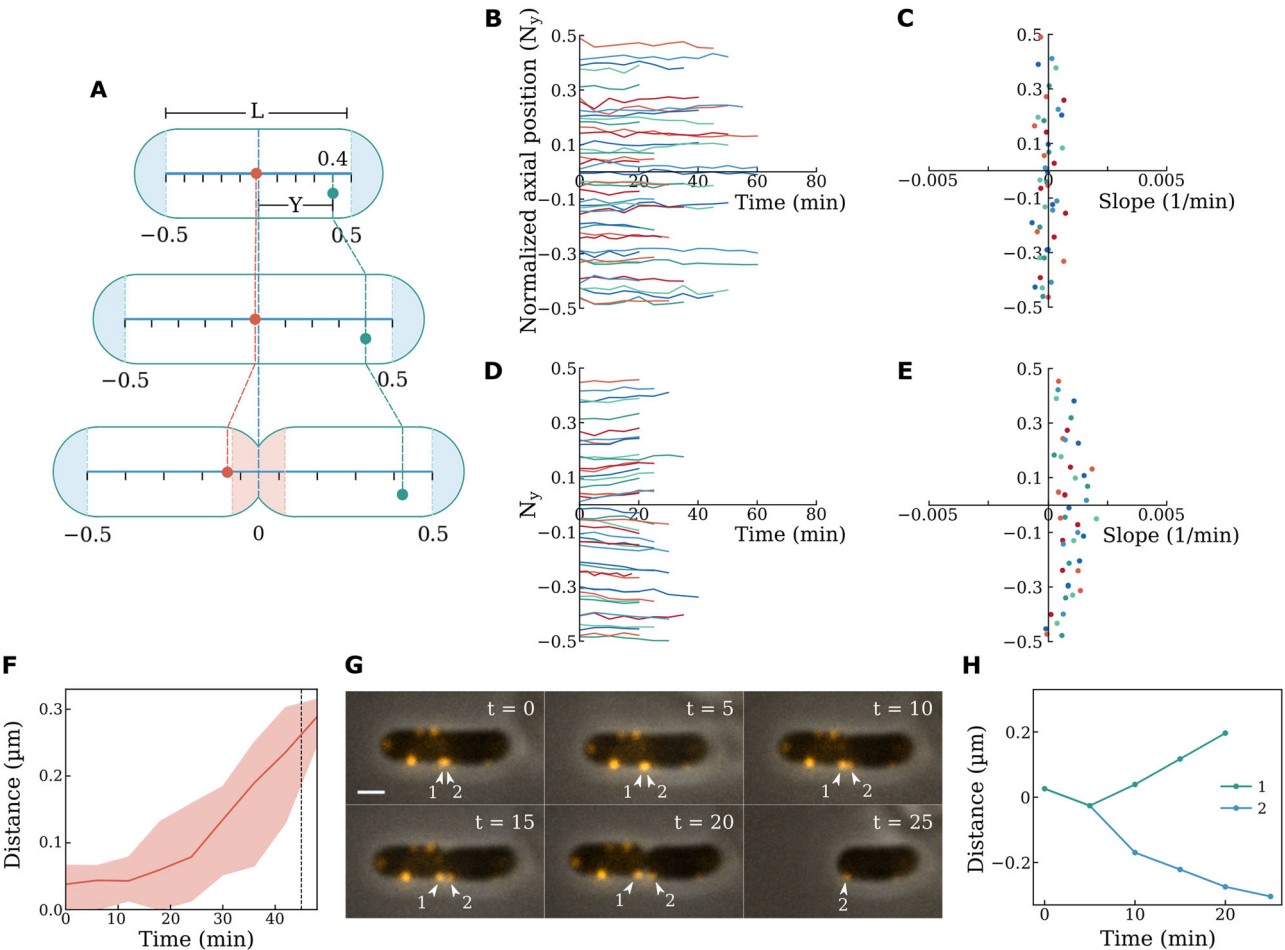

**Fig. 3 Tracing the movement of BFMs in the normalized coordinate. A** A schematic showing changes in the positions of BFMs during cell elongation and division in the normalized coordinate. **B** $N_y$ vs. time of BFMs in the normalized coordinate during cell elongation. All BFMs stayed in the same normalized positions, n = 44 (total number of tracked flagellar motors). **C** Slopes of $N_y$ vs. time during cell elongation [data from (**B**)]. **D** $N_y$ vs. time of BFMs in the normalized coordinate at the beginning of cell division. BFMs moved away from the center of the cell, revealing the formation of a septum (new inert zones), n = 44 (total number of tracked flagellar motors). **E** Slopes of $N_y$ vs. time at the beginning of cell division [data from (**D**)]. **F** Direct tracking of BFMs that were initially present near the center of the cell. The red line shows the average distance and the shaded area covers the upper and lower limits of the data. The dashed line indicates the division time points, n = 6 (total number of tracked flagellar motors). The light color area represents the range of the data. **G** Two BFMs that were initially in close proximity at the center of the cell moved into two separate daughter cells after division. (Scale bar, 1 μm). At least 3 times of the same experiment was repeated with similar results. **H** Trajectories of the two BFMs shown in **G**.

which two BFMs overlapped at the center of the mother cell but later separated into different daughter cells. Of note, after cell division, these BFMs were not located at the tip of the new cell poles, confirming that additional peptidoglycan insertion occurred locally at the center of the cell. Our finding supported the two-phase model in which the cell wall elongation and septal growth are two independent processes[26–28].

**The Bernoulli shift map model.** In summary, we have learned three important facts regarding bacterial cell wall growth by tracing wall-anchored BFMs as landmarks. First, during cell elongation, expansion of the cylindrical cell body is homogeneous in the axial direction. Secondly, the size of the inert zone, where no newly synthesized peptidoglycan is inserted, was defined as 0.27 μm from the tip of the cell poles. Thirdly, after cell division, the new cell poles, which are formed by the local insertion of newly synthesized peptidoglycan, become new inert zones in the daughter cells. Thus, we hypothesized that the partitioning of BFM positions or any cell wall-anchored protein complexes in the active cell wall growth zone across generations can be modeled

using a Bernoulli shift map[29], as detailed by the following iteration equation (Fig. 4A):

$$N_{y(n+1)} = \begin{cases} 2N_{y(n)} - 0.5, & \text{if } 0 < N_{y(n)} \le 0.5 \\ 2N_{y(n)} + 0.5, & \text{if } -0.5 \le N_{y(n)} < 0 \end{cases} \quad (1)$$

where $N_{y(n)}$ and $N_{y(n+1)}$ are the normalized axial position of a BFM in the $n$th and ($n+1$)th generations, respectively.

BFMs located near the center of the cell will be driven away due to active septum growth and will remain at the edge of the new inert zones in the next cell cycle (Fig. 4A, red dot). BFMs sitting on the boundary between the active growth zone and the inert zone will remain in the same position indefinitely (Fig. 4A, blue dot). BFMs in the position of rational numbers will show periodic movements on the Bernoulli shift map; for example, a BFM at $N_{y(1)} = 0.1$ of generation 0 will move to $N_{y(2)} = -0.3$ of generation 1 (Fig. 4A, green dot), then to $N_{y(3)} = -0.1$ of generation 2, then to $N_{y(4)}$ of generation 3, then to $N_{y(5)} = 0.1$ of generation 4 (Fig. 4B, green line).

Subsequently, these model predictions were experimentally tested. In Fig. 4C, we monitored a population of cells going through cell

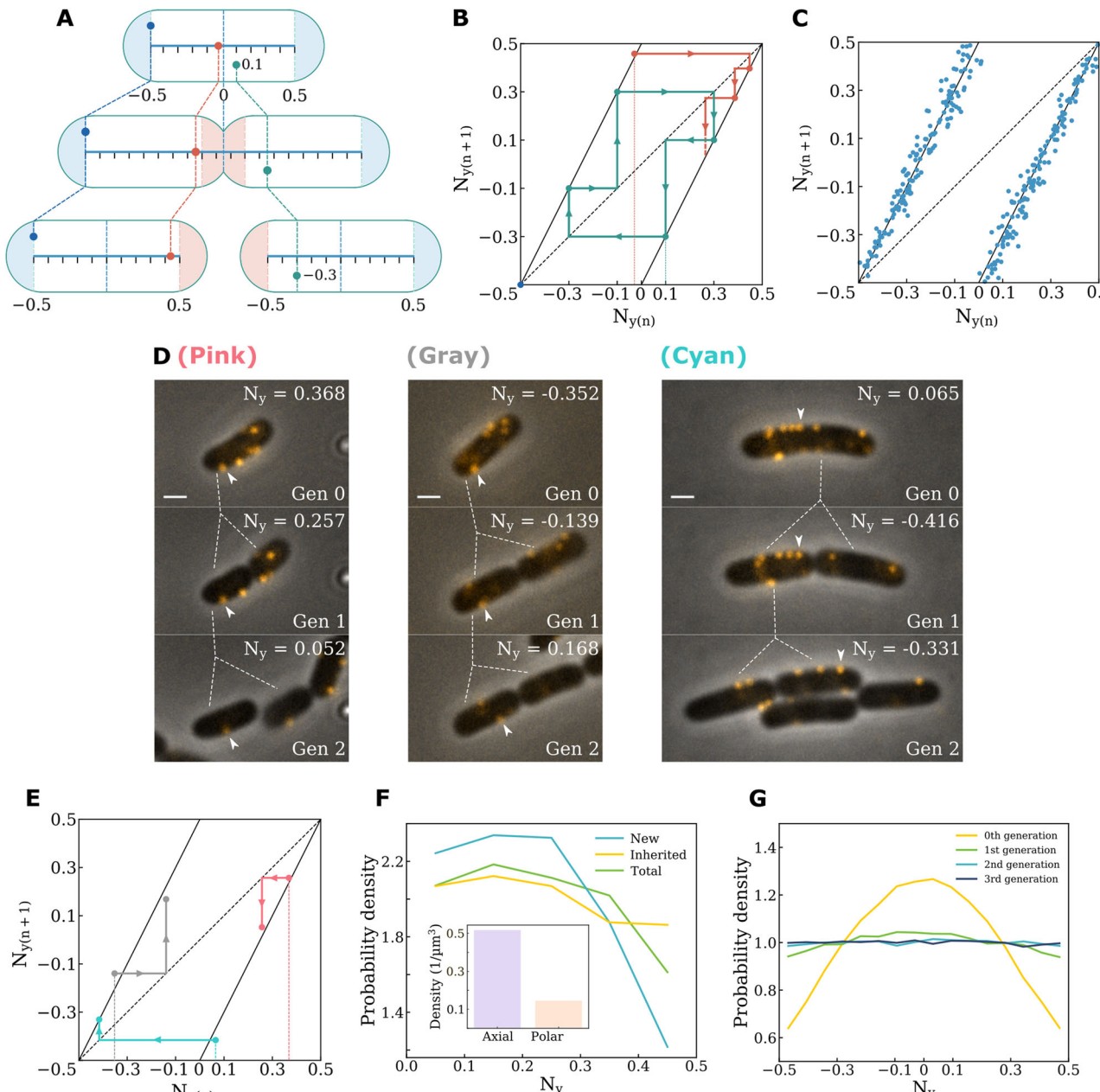

**Fig. 4 Mathematical modeling of BFM positions after cell division. A** A schematic of the Bernoulli shift map model for predicting the partitioning of BFMs across cell generations. Proteins anchored to the central rod region will follow the Bernoulli shift map. **B** Model prediction of the moving trajectories of BFM positions across cell generations on the Bernoulli shift map. $N_{y(n)}$ and $N_{y(n+1)}$ are the normalized axial positions of a BFM in the $n^{th}$ and $(n+1)^{th}$ generations. The dashed line is the guideline for $N_{y(n)} = N_{y(n+1)}$. The black lines are the Bernoulli shift map. The red trajectory is a BFM starting from a position close to the center of the cell; in the next generation, it will move to the edge of the active zone. The green trajectory is a BFM starting from $N_{y(1)} = 0.1$, and moving to $N_{y(2)} = -0.3$, then $N_{y(3)} = -0.1$, then $N_{y(4)} = 0.3$, $N_{y(5)} = 0.1$, completing a cycle. **C** Experimental tracking of BFM positions across two generations (one cell division) of a population of cells (n = 292 BFMs). The data points fell on the Bernoulli shift map. **D** Time-lapse imaging of three representative cases tracking the position of a BFM across three generations. The arrow labels the BFM that were tracked. (Scale bar, 1 μm). **E** Moving trajectories of the three BFMs tracked in **D**. The data points fell on the Bernoulli shift map. **F** The experimental distribution of BFMs in the normalized coordinate along Ny. The green line shows the total BFM distribution by single-color labeling (n = 7757 BFMs), and the yellow and light blue lines show inherited (n = 1508 BFMs) and new (n = 1472 BFMs) BFM distribution by dual-color labeling. Inset: the average BFM density was higher in the axial area than in the polar area. **G** Simulation of BFM distribution evolution over 3 generations. The positions of 200,000 BFMs were assigned following a truncated Gaussian distribution (σ = 0.4). This uneven distribution was smoothed quickly after one generation by the Bernoulli shift map.

division and plotted the relationship between the $N_{y(n)}$ and $N_{y(n+1)}$ of different BFMs. The results show that the positions of BFMs in the $(n+1)^{th}$ vs. $n^{th}$ generation fell on the Bernoulli shift map. In addition, we followed the positions of BFMs in single cells across three generations (Fig. 4D, E), and the moving trajectories of the BFMs also fell on the Bernoulli shift map (Fig. 4E). Therefore, the positions of BFMs, and perhaps other cell wall-anchored protein complexes, across cell generations can be predicted using this model.

**Uneven protein distribution on the central rod can be smoothed after cell division.** Next, we sought to understand why wall-anchored protein complexes follow a Bernoulli shift map to distribute across generations and the biological implications of this distribution. It is generally accepted that BFMs are evenly distributed on the cell envelope of *E. coli*;[30] however, it remains a mystery how this occurs, since no specific binding sites or localization mechanism for BFMs have been identified.

We traced the fluorescently labeled BFMs in the mother cell and their partitioning into daughter cells, but we did not initially study the dynamics of newly produced BFMs in daughter cells following cell division. Given that the total number of BFMs in a cell is the sum of BFMs inherited from the mother cell and newly produced BFMs after cell division, we subsequently performed dual-color fluorescence pulse-chase labeling of BFMs to investigate the evolution of their axial distribution across generations. First, we measured the axial distribution of the total BFMs labeled with Alexa Fluor 488 in mother cells (Fig. 4F, total) and found a non-uniform distribution, with a higher chance of finding a BFM in the center of the cell but a lower chance of finding a BFM near the cell poles; the BFM area density was approximately 4-times lower at the polar cap (Fig. 4F inset).

Subsequently, we traced the partitioning of BFMs into daughter cells and labeled newly produced BFMs with Alexa Fluor 594. Interestingly, we found that the distribution of newly produced BFMs in daughter cells was highly center-biased (Fig. 4F, new), whereas the distribution of inherited BFMs became more uniform (Fig. 4F, inherited). This can be understood as the Bernoulli shift map having a strong smoothing function. Theoretically, a strong center-localized distribution can be smoothed into an even distribution within one generation (Fig. 4G); therefore, the present work provides a physical explanation of BFM positioning dynamics. Any uneven protein distribution on the central rod will be smoothed within a few generations, being fair to the offspring[31]. Moreover, this weak centered BFM distribution can ensure that the cells are more efficient at flagellar bundling[32] and chemotaxis.

## Discussion

There are three important new insights in our Bernoulli shift map model. First, for any model to simulate cell growth across generations, the inert caps must be considered and removed from the length calculation. Second, during cell division, the central endocaps are newly synthesized, which are not part of the original rod. Third, during cell elongation, the expansion of the cylindrical cell body is homogeneous in the axial direction. Therefore, the position of surface anchored proteins such as BFMs will be fixed in the normalized coordinate. After cell division, the BFMs positions will follow the Bernoulli shift map in the normalized coordinate regardless of the cell length. For *E. coli* cell size homeostasis[33], whether it is following a "timer" model, in which cells grow for a specific amount of time before division, or a "sizer" model, in which cells grow to the target size before division, or an "adder" model, in which cells add a constant size between birth and division, as long as the cells divide in the middle, our Bernoulli shift map model will work since it is in the normalized coordinate.

Our work highlights the fact that a simple mathematical model can govern bacterial cell wall expansion, which is the fundamental framework for understanding the partitioning mechanism of cell surface protein complexes during cell elongation and division[34,35]. Achieving an even distribution of cell surface protein complexes through the cell growth and division cycle may be a recurring theme across biological species.

## Methods

**Bacterial strains.** The *E. coli* strain MTB9 derived from RP437 was used in all experiments[22]. The hook gene, *flgE*, was genetically modified to contain an AviTage sequence (GLNDIFEAQKIEWHE) in between I221 and A222 that can be biotinylated by the *E. coli* enzyme, biotin ligase BirA. A plasmid containing the *birA* gene was transformed to enhance the biotinylation efficiency of the hook. The *birA* containing plasmid was constructed from pWR20 plasmid[36] by inserting the *birA* gene in place of the *gfp*, as described before[37], see supplementary information. Strain, plasmid and primers are listed in Supplementary table 1 in the supplementary information.

Bacterial cells used for fluorescence time-lapse microscopy were grown overnight at 37 °C from frozen stocks at a starting dilution of 1:500 in SOC medium (2% bacto-tryptone, 10 mM NaCl, 0.5% yeast extract, 2.5 mM KCl, 10 mM MgCl$_2$, 20 mM glucose) containing 50 µg/mL kanamycin. The overnight culture was diluted 1:100 in SOC medium and regrown at 30 °C to an $OD_{600}$ of ~0.6.

**Flagellar hook labeling.** The 100-µM stock solutions of Alexa Fluor 594 streptavidin (Invitrogen, S11227) and Alexa Fluor 488 streptavidin (Invitrogen, S11223) were prepared by resuspending 1 mg lyophilized powder in DI water. The stocks were stored in 5-µL aliquots at −20 °C.

A 1-mL volume of harvested cells was washed 3 times in motility buffer (MB) (10 mM KPO$_4$, 0.1 mM EDTA) and concentrated to 500 µL, to which 2.5 µL Alexa Fluor 594 streptavidin stock solution was added to give a final concentration of 0.5 µM. Cells were subsequently incubated at room temperature for 25 min in the dark, followed by washing 5 times with 1 mL MB.

For dual-color flagellar motor distribution experiments (Fig. 4F), the existing motors were labeled with Alexa Fluor 488 streptavidin as described above, following which cells were resuspended in 1 mL SOC and incubated at 30 °C for 1 h. After incubation, the newly formed flagellar motors were labeled with Alexa Fluor 594 streptavidin in the same manner.

**Sample preparation.** Cells were immobilized on a poly-L-lysine-coated surface or agarose gel pad for time-lapse observation[38].

The microscope tunnel slides were created by attaching double-sided tape between the microscope slide and cover slip, which had been cleaned in a saturated solution of KOH in 95% ethanol. Poly-L-lysine (Sigma, 0.1%) was added to the tunnel for 1 min to coat a positively charged layer on the glass surface, which was subsequently flushed out with MB. The fluorescently labeled bacterial cells were injected into the tunnel and allowed to settle for 10 min in the dark. Unattached cells were removed using LB (1% bacto-tryptone, 0.5% NaCl, 0.5% yeast extract) for the time-lapse growth experiment and MB for the BFM anchoring stability experiment (Supplementary Fig. 1).

The gel pad was prepared in the center of a gene frame (65 µL, Thermo Scientific, AB0577). For time-lapse observation, 1% low gelling temperature agarose (Sigma, A9414) in LB medium was used. For the flagellar motor distribution experiment, 1% low gelling temperature agarose in MB was used. A 30-µL volume of gel was placed onto the gene frame and a clean coverslip was pressed onto the gel to create a gel pad. After the gel had solidified, the coverslip was removed and a 1-µL aliquot of 10× concentrated cells was added to the gel pad. Finally, the gene frame was sealed with a clean coverslip.

**Microscopy and image acquisition.** The time-lapse phase-contrast and fluorescence imaging for BFM tracking was performed at room temperature on an inverted fluorescence microscope (Nikon, Ti-E) containing the Perfect Focus platform using Nikon NIS-elements AR (version 4.51). Alexa Fluor 594 was excited by a 580-nm LED light source (58 W/cm$^2$) (pE4000, CoolLED, UK) and was observed through a 650-nm emission filter. Alexa Fluor 488 was excited by a 490-nm LED light source (151 W/cm$^2$) and observed through a 530-nm emission filter. Both phase-contrast and fluorescence images were collected by a 1.45 NA ×100 oil objective and captured by an EMCCD camera (Andor, iXon Life 888). Images were captured at 2.5 min, 5 min, or 6 min intervals for 2 h depending on the experimental purpose.

**Image analysis.** All images were analyzed using custom-written Python (3.6.0) scripts and ImageJ (1.51k) (Supplementary Software).

To find the central position of the motors, targeted single-cell image stacks were cropped from the entire fluorescent image stack. Firstly, a Gaussian filter was applied to smooth the image, following which a 12 × 12 pixel (624 × 624 nm) region of interest was cropped based on the local maximum fluorescence intensity. The accurate BFM position was then found by applying 2D Gaussian fitting. To quantitate the position of the motor on the cell, a cell coordinate $(x, y)$ centered in the middle of the cell was defined; $x$ represents the lateral position and $y$ represents the axial position. The BFM position was subsequently the fitted fluorescent BFM position at the center of the cell.

The center of the cell was found as follows. Targeted single-cell image stacks were cropped from the phase-contrast image stack. A Gaussian filter was applied to remove high-frequency noise. The image resolution was further increased 4-times by bicubic interpolation. The bacterial cell contour was obtained using the Otsu thresholding method using outline and skeleton algorithms in ImageJ. For cells that

were just finishing division, a further segmentation process was applied using ImageJ Trainable Weka Segmentation.

Once the cell contour was obtained, the cell midline could be found as follows. Firstly, a custom-written program found the two polar points on the contour, splitting the cell contour in half. Spline interpolation was used to ensure that each half had an equal number of data points with equal spacing. After connecting each point in each half, in order, the midpoint of each connected line was found. The midline is formed by the connection of each midpoint. The center of the midline is then addressed as the center of the cell. All BFM positions then used the center of the cell as a reference point.

**Inert zone size measurement**. To measure the inert zone size, the axial distance between the motor and the nearest pole ($P_y$) was tracked until cell division. Each trace was subjected to linear fitting and the average slope $V_{P_y}$ was obtained. In the $V_{P_y}$ vs. $P_y$ plot (Fig. 1D), the data points were subjected to piecewise linear fitting to obtain the turning point of the $V_{P_y}$:

$$V_{P_y} = \begin{cases} 0, & P_y < P_{yc} \\ k(P_y - P_{yc}), & P_y \geq P_{yc} \end{cases} \quad (2)$$

where $P_{yc}$ is the size of the inert zone. (Supplementary Software).

**Normalized relative axial velocity**. All the $V_{D_y}$ vs. $D_y$ plots were rescaled to a slope equal to 1 and over-plotted in Supplementary Fig. 4A, i.e., plot $V_{D_y}/H$ vs $D_y$.

**Convexity defect detection**. Cell division was detected by finding convexity defects at the center of cells. At each time point, cell contours were used to test for convexity defects. Time-lapse cell images 10 min prior to the detection of convexity defects were used as the cell elongation stage and those following the detection of convexity defects were used as the division stage.

**Simulation of BFM distribution evolution**. First, the positions of 200,000 BFMs were assigned following a truncated Gaussian distribution ($\sigma = 0.4$) centered at $Ny = 0$, and the distribution of the BFMs positions was recorded as "0th generation". Second, the positions of these 200,000 BFMs were used as $Ny(0)$ to calculate $Ny(1)$ according to the Bernoulli shift map Eq. (1). Then we plotted the distribution of $Ny$ (1) in Fig. 4G, which was recorded as "1st generation". Iteratively, $Ny(1)$ of these 200,000 BFMs were used to calculate $Ny(2)$, which was recorded as "2nd generation". Similarly, $Ny(3)$ was generated and recorded as "3rd generation".

**Reporting summary**. Further information on research design is available in the Nature Research Reporting Summary linked to this article.

## Data availability

Data supporting the findings of this work are available within the paper and its Supplementary Information file. All relevant data are available from the corresponding author on request. Source Data are provided within this paper. Source data are provided with this paper.

## Code availability

Custom-written python scripts for imaging and data analysis are provided as Supplementary Software.

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

## Acknowledgements
The authors thank Prof. Richard Berry and Dr. Teuta Pilizota for providing the strains. This work was financially supported by Academia Sinica (AS-TP-106-L04 to Y.LS. and AS-TP-106-L04-3 to C.J.L.), the Ministry of Science and Technology, Republic of China, under contract No. MOST-107-2112-M-008-025-MY3 and MOST-109-2628-M-008-001-MY4 to C.J.L., the National Natural Science Foundation of China (No. 31722003 and 31770925) to F.B.

## Author contributions
C.J.L., F.B., Y.L.S., and Y.J.S. designed the project; Y.J.S., A.C.L., X.Y.Z., T.S.L, Y.C.S., and C.J.L. performed experiments; Y.J.S., F.B., Y.L.S., Y.C.S., and C.J.L. analyzed data; C.J.L., F.B., and Y.J.S. model the data; C.J.L. supervised the project; C.J.L., F.B., and Y.L.S. wrote the paper with input from all authors.

## Competing interests
The authors declare no competing interests.
