## [Peer Review File · Nature Communications]

REVIEWER COMMENTS

Reviewer #1 (Remarks to the Author):

Sun et al develop an interesting experimental approach to dynamically track bacterial cell wall growth by following the movement of fluorescently labelled peptidoglycan anchored flagellar motors. The method is used to reveal the regions of inert and active growth zones in *E. coli*, and how growth is distributed along the bacterial cell surface during elongation and septation. While the experimental method is useful the mathematical analysis and the model proposed raise several questions that the authors should address (see below). Firstly, it is not clear how the growth rate inferred from the movement of flagellar motors correlate with growth rate measured from cell size elongation. Second, it is unclear how the findings correlate with previously studied models for peptidoglycan insertion. Furthermore, the authors propose a 'Bernoulli shift map' which is not properly tested with the available experimental data. Methods for simulation and data analysis need to be presented in detail. Finally, it will be useful to test the approach in different growth conditions to understand what geometric factors regulates the regions of inert and active growth.

Specific Comments:

- It would be informative to know how growth conditions modulate the position of the boundary between the active and inactive zones changes, e.g. with the quality of nutrients. Is there a simple geometric relationship between the cell diameter and the position of the boundary?

- Should the x-label in Fig. 1D read "initial axial position P_{y0} " as mentioned in the main text? Relatedly, what would the plot of instantaneous velocity vs instantaneous axial position look like? What does the 'n' number indicate – data pooled from different cells?

- The growth rate of a single cell is considered to be constant during the cell cycle (Wang et al. *Curr Biol* 2010, Tanouchi et al., *Nature* 2015). How do you reconcile this with the increase in axial growth rate reported in Fig. 2D? Is this finding consistent with growth rate inferred from cell size measurements (for example calculating the changes in length of the cell from the cell contours obtained from phase contrast images)?

- Could the increase in relative axial velocity be a result of relative movement of BFM's? In the cited paper (Darnton and Berg, *J bacterial* 2008), the extraction of flagellar filaments by pulling failed, which led the authors to claim that the flagella are firmly anchored. However, this does not necessarily imply that during cell elongation the position is fixed. There could be relative movement between pairs of BFM's.

This question was addressed by arresting growth and observing invariance in the position of BFM's in time. Can the arrest of growth anchor the BFM's? For example, if there was relative motion of the BFM's with new peptidoglycan insertion, when the insertion is arrested, the BFM's would keep their position as no new cell wall material is synthesized.

- The axial relative velocity is calculated as $V_{Dy} = \frac{D_y}{t}$ from which is found that V_{Dy} increases linearly with D_y and then the growth is described as $V_{Dy} = HD_y$, where H indicates the instantaneous growth rate per unit distance and H increases as cell elongates. In the supplement it is mentioned that "all the V_{Dy} vs D_y plots were rescaled to a slope equal to 1 and over-plotted in Fig. 3E". Firstly, the figure reference should be to Fig. 2E. Secondly, given the way H is defined, it is obvious that the data in Fig. 2E should collapse to a straight line of slope 1. The order in which these new quantities are introduced is confusing.

- Fig. 3F shows the tracking of BFM's near the center of the cell. Please clarify whether these measurements are consistent with the situation where you have 2 BFM's around the middle of the cell that are split into the future daughter cells (each BFM on different daughter cells) by measuring the velocity between these 2 BFM's? (similar to Fig. 2 rationale)

- Fig. 3 supports a model for septal growth. Please discuss how your findings relate to previously introduced model for septal growth (Reshes et al *Biophys J* 2008, Banerjee et al *Nat Microbiol*

2017)

- Fig. 4 – the Bernoulli shift map predicts cyclicity in the normalized positions. Can you support this observation with experimental data? For example, in Fig. 4E you could follow the BFM's for more generations to observe such cyclic patterns.

- Fig. 4c is an obvious result given how the normalized position is defined, and it is not a test of the model. Since a cell symmetrically divides, the normalized position is bound to fall on one of the lines. A better test of the model would be comparing N_y trajectories with experimental data.

- It is not clear where the results in Fig. 4G come from. Please detail how you perform the simulations, when do you introduce a smoothing function in the Bernoulli shift, what is the smoothing function etc.

Reviewer #2 (Remarks to the Author):

The paper "Probing bacterial cell wall growth by tracing wall-anchored protein complexes" by Sun et. al. uses what is claimed to be a novel technique, measuring cell growth by labeling flagellar motors and tracking them microscopically. Using this technique, they state 3 findings in regards to cell wall growth of rod-shaped bacteria:

1. That cell growth occurs by the insertion of material throughout the rod in a non-twisting fashion, in disagreement with previous observations.
2. That the cell caps are inert, and that "for the first time, the boundary between the active zones for cell wall growth in live *E. coli* has been determined."
3. That the new poles become inert caps.

Next, in the last figure of the paper, they examine the distribution of flagellar motors and how it arises. In this short section, they find that the flagellar motors are distributed non-randomly and that they are center-biased.

While the technique is interesting, and they back these findings with models, the problem is that the large majority of this work is not novel. The main novel part of the work is confined in one figure, which is not well flushed out. Thus I cannot recommend this paper to be published in Nature Communications, as this work lacks sufficient new results to be in this journal, which normally publishes articles I have great respect for. Rather, as this paper does indeed contain a novel technique and nice mathematical modeling, I suggest they submit this to a more appropriate journal, like Biophysical Society. To explain:

First, all of their results in regards to cell growth are long-established findings, well-accepted in the field. In fact, the uniform insertion has been well-studied for over 2 decades. The same publication, and others, describe the caps are inert, measure their size, and describe how they arise. But in this work, these findings are stated as novel, or "proof" to lend evidence to a given model. However, the reality is all of these findings are taken as ground truth in the field. References supporting this conclusion are appended to the end of this review.

Second, the technique that is presented as novel has already published work by Wang et al. (2011), showing opposing results than here in this study, where they observed twisting during growth, a discrepancy mentioned but not discussed in this paper. That being said, I do appreciate the rigor and uses of the technique for more than just showing twisting: it is used to qualitatively and quantitatively describe the dynamics of wall material insertion during growth, These dynamics have been previously explored with other techniques, as mentioned above, which, however, have

some limitations compared to the technique presented here. Notably, other techniques, like fluorescent D-amino acids are more perturbative of wall synthesis and their effects on the dynamics of wall properties have not yet been fully elucidated.

Thus, the only novel finding in this work is how the flagellar motor distribution arises. This is indeed an interesting result, where they find that while the newly produced flagella are non-uniform (center-biased), they become more uniform as they are inherited. In this part of the paper, we can see another advantage of the technique used, which is the high spatial resolution of the labeled flagella in showing wall material inheritance. This allowed the authors to show how flagella that were within spatial resolution limits in one mother cell, were split in two in the daughter cells, an interesting result that can be further explored. While this is a nice result, this is only one panel (and finding) out of four in this paper that is novel. I am confident that, if further fleshed out, studied, and explained, this result itself could stand on its own in a biological journal.

But currently, it is my overall opinion that this work, since it developed a new technique and did some advanced tracking and modeling on well-studied systems (thus verifying the technique) fits better in a more technique or physics-oriented journal, where such very well done quantitation and modeling, on already described systems, would be better appreciated.

References:

1. Varma, A., Pedro, M. A. de & Young, K. D. FtsZ Directs a Second Mode of Peptidoglycan Synthesis in *Escherichia coli*. *J Bacteriol* 189, 5692–5704 (2007).
2. Pedro, M. A. D., Schwarz, H. & Koch, A. L. Patchiness of murein insertion into the sidewall of *Escherichia coli*. *Microbiology* 149, 1753–1761 (2003).
3. Pedro, M. A. de, Quintela, J. C., Höltje, J. V. & Schwarz, H. Murein segregation in *Escherichia coli*. *J Bacteriol* 179, 2823–2834 (1997).
4. Ursell, T. S. et al. Rod-like bacterial shape is maintained by feedback between cell curvature and cytoskeletal localization. *Proc National Acad Sci* 111, E1025–E1034 (2014).

Reviewer #3 (Remarks to the Author):

Regulating the dynamics of bacterial cell growth is highly important as random growth would result in aberrant cell morphology, loss of cell functions or, eventually, inability of division and cell death. However, how and where cells insert new cell-wall material to maintain cell shape and function during growth is still somewhat unclear. One of the problems is that most methods used for identifying the corresponding loci within the cell envelope relies on using labeled D-amino acids, which may not allow appropriate spatiotemporal resolution for the analysis. In this study Sun and coworkers used a clever alternative approach by labeling cell envelope complexes that remain static within the cell envelope. To this end, they used the hooks of the flagellar systems in *Escherichia coli* for fluorescent labeling, which are distributed around the cell body in a so-called peritrichous flagellation pattern while excluding the cell poles. Once the outer-cell structures, such as the hook are formed, the flagella have been demonstrated to remain static and only shift their general position with respect to the cell and/or other flagella complexes upon cell growth. Thus, the shift of flagellar complexes position can be used to indirectly determine the approximate position of cell-elongation areas and to describe the growth behavior.

The authors used this set-up to determine, for the first time, the boundary between the inert and active zones for cell wall growth in *E. coli*, which locates – under the conditions tested – 0.27 μm from the tip of the cell poles. The results further indicated homogenous growth in the axial direction but not the lateral direction so that the diameter of the cell body does not change and not twisting occurs, which fits well to observations made in previous studies. Further growth only

occurs at mid-cell upon formation of a new septum upon cell division, so that even flagellar complexes directly localized at mid-cell will not become polar after cell fission. These zones, after cell division, become the new inert zones of the daughter cells. The data obtained were used to formulate a Bernoulli shift map that was employed to predict partitioning of cell wall-anchored complexes (such as the flagella) after cell division in *E. coli*, which was then challenged experimentally. Newly produced flagella appeared to be center biased, which would allow an even distribution of flagella within one or more generations among the daughter cells.

In my opinion, the authors used a clever approach to determine cell growth dynamics in *E. coli*, which may now be challenged by studies on other cell species. The authors should mention that, for the time being, this approach is limited to *E. coli* (in particular in the abstract, line 22). Apart from this, I have no issues with this manuscript.

Response to reviewers' comments

Reviewer #1 (Remarks to the Author):

Sun et al develop an interesting experimental approach to dynamically track bacterial cell wall growth by following the movement of fluorescently labelled peptidoglycan anchored flagellar motors. The method is used to reveal the regions of inert and active growth zones in E. coli, and how growth is distributed along the bacterial cell surface during elongation and septation. While the experimental method is useful the mathematical analysis and the model proposed raise several questions that the authors should address (see below). Firstly, it is not clear how the growth rate inferred from the movement of flagellar motors correlate with growth rate measured from cell size elongation. Second, it is unclear how the findings correlate with previously studied models for peptidoglycan insertion. Furthermore, the authors propose a 'Bernoulli shift map' which is not properly tested with the available experimental data. Methods for simulation and data analysis need to be presented in detail. Finally, it will be useful to test the approach in different growth conditions to understand what geometric factors regulates the regions of inert and active growth.

Response: We thank Reviewer #1 for the nice summary of our work and all the positive comments.

Specific Comments:

- It would be informative to know how growth conditions modulate the position of the boundary between the active and inactive zones changes, e.g. with the quality of nutrients. Is there a simple geometric relationship between the cell diameter and the position of the boundary?

Response: We thank Reviewer #1 for this suggestion. In our original manuscript, we presented BFM tracking in active growing cells and measured the axial position of BFMs from their nearest cell pole (P_y). From the average velocity of BFMs, we can identify the boundary between the active and inert zones for peptidoglycan insertion. In LB medium, the boundary was found to be located at the axial position $P_{y0} = 0.27 \pm 0.04 \mu\text{m}$.

In our revision, we have tried to re-determine the boundary position separating the active and inert zones when *E. coli* cells were grown in TB medium (1% bacto-tryptone, 0.5% NaCl), a less rich growth medium. As shown in **Response Fig. 1**, the axial position of the boundary was determined to be located at $P_{y0} = 0.22 \pm 0.03 \mu\text{m}$. The inert zone was found to slightly reduce its size whereas the cell diameter was almost the same for cells grown in TB and LB ($D_{\text{TB}}: 0.51 \pm 0.03 \mu\text{m}$; $D_{\text{LB}}: 0.51 \pm 0.02 \mu\text{m}$).

A systematic investigation exploring all combinations of nutrients in the growth buffer requires a substantial amount of time, and we believe it is beyond the scope of our current research. We hope to leave the full investigation of this interesting suggestion to a future work.

Response Fig. 1 Real-time monitoring of *E. coli* cell wall growth using fluorescently labeled flagellar motors (BFMs) as landmarks. (A) Sample traces tracking the axial positions (P_y) of BFMs during cell elongation in TB buffer, $n = 22$. (B) Average velocity of BFM movement during cell growth identified an inert zone near the cell poles. Green dots: BFMs in the inert zone; blue dots: BFMs in the active zone, $n = 56$ (total number of tracked flagellar motors).

- Should the x-label in Fig. 1D read “initial axial position P_{y0} ” as mentioned in the main text? Relatedly, what would the plot of instantaneous velocity vs instantaneous axial position look like? What does the ‘n’ number indicate – data pooled from different cells?

Response: We thank Reviewer #1 for pointing out this mistake. Yes, the x-label in Fig. 1D should read “Initial axial position (P_{y0}) from the nearest pole (μm)”. We have corrected this mistake.

Per Reviewer #1’s suggestion, we have plotted the instantaneous velocity vs. instantaneous axial position in **Response Fig. 2**, which looks very similar to **Fig. 1D**. The instantaneous velocity also increases when the cell is longer.

We thank Review #1 for pointing the omitted definition of ‘n’ number in our manuscript. In fact, ‘n’ represents the total number of tracked flagellar motors in Fig. 1D. We have added descriptions of each “n” in the figure legends.

Response Fig. 2 The relationship between instantaneous velocity of BFM movement during cell growth. The measurement time interval is 25 mins. Green dots: 0-25 mins; blue dots: 25-50 mins; red dots: 50-75 mins.

- *The growth rate of a single cell is considered to be constant during the cell cycle (Wang et al. Curr Biol 2010, Tanouchi et al., Nature 2015). How do you reconcile this with the increase in axial growth rate reported in Fig. 2D? Is this finding consistent with growth rate inferred from cell size measurements (for example calculating the changes in length of the cell from the cell contours obtained from phase contrast images)?*

Response: This is an interesting point. The conclusion that the growth rate of a single bacterial cell is constant during the cell cycle was drawn primarily from the two papers that Reviewer #1 has mentioned (Wang et al. Curr Biol 2010, Tanouchi et al., Nature 2015). However, when we had a closer look at the data, we reached a different conclusion.

In **Response Fig.3 (A)** we re-analyzed the single cell growth data presented in Tanouchi et al. 2015 (the raw data was published on Scientific Data 2017 and we have downloaded the data), we noticed that the growth rate of a single bacterial cell is not always constant. First, we see that in the semi-log scale, the relationship between $\ln(\text{Length})$ and time is not strictly a simple linear line. To further clarify, we performed a three-sections growth rate fitting to one growth cycle, as shown in **Response Fig. 3 (B)**, to obtain the cell growth rates at different timings of one growth cycle. The histograms of the three-sections growth rates surveyed over many growth cycles are shown in **Response Fig.3 (C)**, which clearly demonstrates an increase in cell growth rate during a bacterial growth cycle.

Response Fig. 3 Re-analysis of the single cell growth measurement from Tanouchi et al., Scientific Data 2017. (A) The relationship between cell length and time plotted in semi-log scale. (B) A sample trace with three-sections linear fitting. (C) Histograms of three-sections growth rates surveyed from a total of 4,550 growth cycles from 65 cells. The fitted growth rates are increasing during the cell growth cycle.

To further confirm the growth rate from the cell contour data in our images, we perform the same analysis. First the relationship between $\ln(\text{Length})$ and time is not a simply linear line as growth rate increase in three sections growth rate fitting, as shown in **Response Fig. 4 (A)**. The average growth rate from three-section growth rate surveyed of 14 cells (14 generations) are shown in **Response Fig. 4 (B)**, which also clearly demonstrates an increase in cell growth rate during a bacterial growth cycle.

Response Fig. 4. Analysis of the single cell growth measurement from our data. (A) A sample trace with three-sections linear fitting. The relationship between cell length and time plotted in semi-log scale. (B) Average and standard deviation of three-sections growth rates surveyed from a total of 14 growth cycles from 14 cells. The fitted growth rates are increasing during the cell growth cycle.

- Could the increase in relative axial velocity be a result of relative movement of BFMs? In the cited paper (Darnton and Berg, *J bacterial* 2008), the extraction of flagellar filaments by pulling failed, which led the authors to claim that the flagella are firmly anchored. However, this does not necessarily imply that during cell elongation the position is fixed. There could be relative movement between pairs of BFMs.

This question was addressed by arresting growth and observing invariance in the position of BFMs in time. Can the arrest of growth anchor the BFMs? For example, if there was relative motion of the BFMs with new peptidoglycan insertion, when the insertion is arrested, the BFMs would keep their position as no new cell wall material is synthesized.

Response: We thank Reviewer #1 for asking this critical question. A basic assumption of our work is that BFMs are firmly anchored to the cell wall, and any relative movement between pairs of BFMs is resulted from cell wall insertion and growth. We provide a series of experimental evidence to support this hypothesis:

1) Evidence 1

As shown by Darnton NC and Berg HC (*J. Bacteriol.* 2008), they tried to extract flagellar filaments from live *Salmonella enterica* in motility buffer by pulling on them with an optical trap but failed, even when they used forces large enough to straighten the filaments. Thus, they concluded that flagella are firmly anchored to the cell wall when cells are not growing.

2) Evidence 2

By monitoring the change in location of fluorescently labeled bacterial flagellar hooks, we did not observe any movement beyond experimental error (**Fig. S1**). Therefore, we concluded that flagella are firmly anchored to the cell wall when cells are not growing.

3) Evidence 3

We monitored the rotation of the cell body in a tethered cell experiment when the cell was slowly growing in growth medium. We found that both the rotation speed and geometry of the cell body driven by a BFM were very stable, whilst the cell body was actually growing (**Response Fig. 5A-B**). This observation strongly suggests that the BFM is firmly anchored to the cell wall, otherwise it cannot stably output a rotational torque to drive the cell body rotation.

4) Evidence 4

Also, in a tethered cell experiment conducted in growth medium, we applied a transient hydrodynamic flow to push the cell body away from the original rotation geometry. When the flow was stopped, we found that the rotation speed and geometry of the cell body were restored and the position of the BFM that drove the rotation of the cell body was almost identical to the position before the flow was applied (**Response Fig. 5C**). This again strongly suggests that the BFM is firmly anchored to the cell wall.

Based on the above evidence, we believe BFMs are firmly attached to the cell wall even when the cell is actively growing, and the assumption of our work is scientifically sound.

Response Fig. 5 Additional experimental results to support that BFMs are firmly anchored to the cell wall during cell elongation. (A) A tethered cell experiment showing stable rotation of *E. coli* cell body in growth medium. (B) The same cell as in (A) showing stable rotation of cell body after 20 mins in growth medium while the cell elongates. (C) A tethered cell rotated by a BFM and the hook of the BFM was fluorescently labelled. The coordinate of the BFM from the cell pole was $[226\text{nm}, 1160\text{nm}]$. After a strong hydrodynamic flow, the BFM position from the cell pole was $[225\text{nm}, 1135\text{nm}]$.

- The axial relative velocity is calculated as $V_{Dy} = \frac{D_y}{t}$ from which is found that V_{Dy} increases linearly with D_y and then the growth is described as $V_{Dy} = HD_y$, where H indicates the instantaneous growth rate per unit distance and H increases as cell elongates. In the supplement it is mentioned that "all the V_{Dy} vs D_y plots were rescaled to a slope equal to 1 and over-plotted in Fig. 3E". Firstly, the figure reference should be to Fig. 2E. Secondly, given the

way H is defined, it is obvious that the data in Fig. 2E should collapse to a straight line of slope 1. The order in which these new quantities are introduced is confusing.

Response: We thank Reviewer #1 for spotting this typo and we have corrected it. Yes, if all of the individual curves in Fig. 2C are linear, they should collapse to a straight line of slope 1. That's exactly what we were trying to present to demonstrate that $V_{\{Dy\}}=HD_y$ holds true at all stages.

- Fig. 3F shows the tracking of BFM's near the center of the cell. Please clarify whether these measurements are consistent with the situation where you have 2 BFM's around the middle of the cell that are split into the future daughter cells (each BFM on different daughter cells) by measuring the velocity between these 2 BFM's? (similar to Fig. 2 rationale)

Response: Yes, these measurements are consistent.

Fig. 3F and 3H are consistent as all of these BFM's were moving away from the cell center. Fig. 3H represents a special case in which the initial distance between the two BFM's are within the optical diffraction limit (<200 nm). This case suggests that the septal growth initializes in a narrow region between the two BFM's and subsequently pushes the two BFM's away.

To repeat the analysis used in Fig. 2, we need to have at least three fluorescent BFM's on the same cell. Therefore, we cannot perform the same analysis with the data in Fig. 3G and Fig. 3H.

We then calculated the relative velocity between these 2 BFM's in Figure 3H (**Response Fig. 6**).

Response Fig. 6 The relative velocity between the 2 BFM's in Fig. 3H.

- Fig. 3 supports a model for septal growth. Please discuss how your findings relate to previously introduced model for septal growth (Reshes et al Biophys J 2008, Banerjee et al Nat Microbiol 2017)

Response: We thank Reviewer #1 for this suggestion. Our data support the two-phase model in which the cell wall elongation and septal growth are two independent processes. We have added a sentence discussing this (Line 147).

- Fig. 4 – the Bernoulli shift map predicts cyclicity in the normalized positions. Can you support this observation with experimental data? For example, in Fig. 4E you could follow the BFM's for more generations to observe such cyclic patterns.

Response: We thank Reviewer #1 for asking this question. In a Bernoulli shift map, ideally, for any BFM position of a rational number, it will move cyclically in the normalized position. Fig. 4B is a theoretical prediction of the cyclic travel trajectories of two representative BFMs across cell generations. We have to clarify that Fig. 4C is in fact an experimental validation of the Bernoulli shift map. In Fig. 4C we recorded $N_{y(n)}$ and $N_{y(n+1)}$ in the normalized coordinates of a population of cells ($n = 292$ BFMs) across two generations (one cell division), and we saw the blue dots ($N_{y(n)}, N_{y(n+1)}$) all fell on the Bernoulli shift map. In Fig. 4D and Fig. 4E, we managed to track the BFM's position across more than two generations and the movement of the three representative cases all followed the Bernoulli shift map. Unfortunately, in our experimental setup, it was very hard to track the BFM's position continuously across many generations because cells were overlapping at a high cell density condition, which made an accurate measurement of BFM's position difficult.

- Fig. 4c is an obvious result given how the normalized position is defined, and it is not a test of the model. Since a cell symmetrically divides, the normalized position is bound to fall on one of the lines. A better test of the model would be comparing N_y trajectories with experimental data.

Response: We hope to clarify the potential misunderstanding here. Fig. 4C is in fact an experimental validation of the Bernoulli shift map. In Fig. 4C we recorded $N_{y(n)}$ and $N_{y(n+1)}$ in the normalized coordinates of a population of cells ($n = 292$ BFMs) across two generations (one cell division), and we saw the blue dots ($N_{y(n)}, N_{y(n+1)}$) all fell on the Bernoulli shift map.

- It is not clear where the results in Fig. 4G come from. Please detail how you perform the simulations, when do you introduce a smoothing function in the Bernoulli shift, what is the smoothing function etc.

Response: We thank Reviewer #1 for pointing this out. We did not introduce any 'smooth function' when performed the simulation. What we actually mean is that the Bernoulli shift map itself can act as a smooth function. Theoretically, a strong center-localized distribution can be smoothed into an even distribution within one generation (Fig. 4G); therefore, the present work provides a physical explanation of BFM positioning dynamics. To follow the Bernoulli shift map when partitioning cell wall anchored protein complexes across generations has one biological advantage: any uneven protein distribution on the central rod will be smoothed within a few generations, being fair to the offspring.

We have also provided the details of how we performed the simulation. (Line 299)

'First, the positions of 200,000 BFMs were assigned following a truncated Gaussian distribution ($\sigma = 0.4$) centered at $N_y = 0$, and the distribution of the BFMs positions was recorded as '0th generation'. Second, the positions of these 200,000 BFMs were used as $N_{y(0)}$ to calculate $N_{y(1)}$ according to the Bernoulli shift map equation [1]. Then we plotted the distribution of $N_{y(1)}$ in Fig. 4G, which was recorded as '1st generation'. Iteratively, $N_{y(1)}$ of these 200,000 BFMs were used to calculate $N_{y(2)}$, which was recorded as '2nd generation'. Similarly, $N_{y(3)}$ was generated and recorded as '3rd generation'.'

Reviewer #2 (Remarks to the Author):

The paper “Probing bacterial cell wall growth by tracing wall-anchored protein complexes” by Sun et. al. uses what is claimed to be a novel technique, measuring cell growth by labeling flagellar motors and tracking them microscopically. Using this technique, they state 3 findings in regards to cell wall growth of rod-shaped bacteria:

- 1. That cell growth occurs by the insertion of material throughout the rod in a non-twisting fashion, in disagreement with previous observations.*
- 2. That the cell caps are inert, and that “for the first time, the boundary between the active zones for cell wall growth in live E. coli has been determined.”*
- 3. That the new poles become inert caps.*

Next, in the last figure of the paper, they examine the distribution of flagellar motors and how it arises. In this short section, they find that the flagellar motors are distributed non-randomly and that they are center-biased.

While the technique is interesting, and they back these findings with models, the problem is that the large majority of this work is not novel. The main novel part of the work is confined in one figure, which is not well flushed out. Thus I cannot recommend this paper to be published in Nature Communications, as this work lacks sufficient new results to be in this journal, which normally publishes articles I have great respect for. Rather, as this paper does indeed contain a novel technique and nice mathematical modeling, I suggest they submit this to a more appropriate journal, like Biophysical Society. To explain:

Response: We thank Reviewer #2 for the summary of our work. With all due respect, we cannot fully agree with Reviewer #2’s comments. We have toned down some of our statements in response to Reviewer #2’s criticism. However, for those comments we cannot agree, we provide our argument and explanation as detailed below.

First, all of their results in regards to cell growth are long-established findings, well-accepted in the field. In fact, the uniform insertion has been well-studied for over 2 decades. The same publication, and others, describe the caps are inert, measure their size, and describe how they arise. But in this work, these findings are stated as novel, or “proof” to lend evidence to a given model. However, the reality is all of these findings are taken as ground truth in the field. References supporting this conclusion are appended to the end of this review.

Response: We thank Reviewer #2 for leading our attention to those published works. We are aware of many of the previous literatures. Particularly,
Pedro, M. A. de, Quintela, J. C., Höltje, J. V. & Schwarz, H. Murein segregation in *Escherichia coli*. J Bacteriol 179, 2823–2834 (1997).

is considered as a truly innovative research work which pioneered the study of murein segregation and bacterial cell wall growth.

In the references provided by Reviewer #2, it is true that some conclusions of our manuscript have been qualitatively established using the label-and-chase strategy of D-cysteine accompanied by immunofluorescence or immunoelectron microscopy visualization. We fully agree with this, but we want to argue that even within this model framework, our study used a novel strategy to probe the dynamics of bacterial cell wall growth with high spatial and temporal resolution in live cells, which has contributed many novel insights into bacterial cell wall growth.

1) The research strategy we presented in this work is very innovative.

In this work, we probed the dynamics of bacterial cell wall growth by tracing the movement of the peptidoglycan-anchored protein complexes, bacterial flagellar motors (BFMs), during cell growth and division in *E. coli*. By measuring the changes in distance between any pair of these fluorescent 'landmarks' or between a fluorescent BFM and a cell pole using time-lapse microscopy, we investigated the dynamic pattern of murein insertion with high spatial and temporal resolution, while the natural process of bacterial cell wall assembly was not perturbed.

The previously used D-cysteine labeling was for sure a milestone in studying bacterial cell growth. However, the method has several limitations. Reviewer #2 also acknowledged in his/her comments below '*These dynamics have been previously explored with other techniques, as mentioned above, which, however, have some limitations compared to the technique presented here. Notably, other techniques, like fluorescent D-amino acids are more perturbative of wall synthesis and their effects on the dynamics of wall properties have not yet been fully elucidated.*'

2) The quantitative aspect of our observations is very novel.

We agree with Reviewer #2 that the qualitative picture of bacterial cell wall growth was well established. However, we want to emphasize that because of the high spatial and temporal resolution enabled by our research strategy, we were able to resolve many quantitative details of bacterial cell wall growth.

For instance, though it has been previously shown that the two caps of *E. coli* are the inert zones for peptidoglycan insertion, the location of the boundary that separates the active and inert zones remains unknown. In this work, we provided solid evidence showing that the boundary between the inert and active zones for cell wall growth in live *E. coli* is located at $Py0 = 0.27 \pm 0.04 \mu\text{m}$ from the cell pole.

Though it has been suggested that murein insert uniformly over the cylindrical sidewall region of *E. coli*, in this work we provided quantitative evidence to reveal the dependence between the relative velocity (V_{Dx} , V_{Dy}) and relative distance (D_x , D_y) between paired BFMs. We found that V_{Dy} increased linearly with increases in D_y ($V_{Dy} = H D_y$, where H is the 'Hubble's parameter' for bacterial cell wall expansion), indicating a homogeneous cell wall growth pattern in the axial direction. Further, we

observed $V_{Dx} = 0$ in the lateral direction, indicating cell wall growth only occurs in the axial direction in *E. coli* without increasing the diameter or twisting of the cell body, an observation that differs from previous reports.

Most importantly, because of the high spatial and temporal resolution enabled by our research strategy, we were able to track the partitioning of BFM positions across generations and for the first time, we found that the partitioning of BFMs or any cell wall-anchored protein complexes in the active cell wall growth zone across generations follows a Bernoulli shift map.

3) We have credited (or will fully credit) the contributions from other researchers.

We must clarify that we did not intend to ignore previous knowledge or contributions from other researchers.

In the Introduction part of our manuscript, we clearly stated '*Previous studies have taken advantage of the fact that bacteria incorporate fluorescent D-amino acids into the cell wall during synthesis, demonstrating qualitative measurement of cell wall insertion patterns (7, 15, 16).*'

If Reviewer #2 thinks here needs more citations, we are happy to cite the references provided by Reviewer #2.

On the first conclusion of our work regarding the boundary position between active and inert zones, we have stated '*During cell growth, the two caps remain intact and the cylinder elongates. Presumably, the hemispherical caps are the inert zones and the cylindrical body is the active zone for peptidoglycan insertion (7, 14).*'

Here, citation 7 is the classic paper

7. M. A. DePedro, J. C. Quintela, J. V. Höltje, H. Schwarz, Murein segregation in *Escherichia coli*. *J. Bacteriol.* 179, 2823–2834 (1997).

If Reviewer #2 thinks here needs more citations, we are happy to cite the references provided by Reviewer #2.

On the second conclusion of our work regarding the uniform insertion, we have changed to '*Interestingly, we found that V_{Dy} increased linearly with increases in D_y , indicating a homogeneous cell wall growth pattern in the axial direction (Fig. 2C), consistent with previous observations using D-amino acid (7, 23).*'

23. Pedro, M. A. D., Schwarz, H. & Koch, A. L. Patchiness of murein insertion into the sidewall of *Escherichia coli*. *Microbiology+* 149, 1753–1761 (2003).

On the third conclusion of our work regarding the emergence of septum during cell division, we have added more citations to previous works

'As the cell entered the division stage (when a convexity defect began to emerge at the center of the cell), the insertion of newly synthesized peptidoglycan was no longer homogeneous in the axial direction within the active zone due to the formation and constrictive nature of the division septum (7, 14).'

'Of note, after cell division, these BFM's were not located at the tip of the new cell poles, confirming that additional peptidoglycan insertion occurred locally at the center of the cell. Our finding supported the two-phase model in which the cell wall elongation and septal growth are two independent processes (26-28).'

26. Varma, A., Pedro, M. A. de & Young, K. D. FtsZ Directs a Second Mode of Peptidoglycan Synthesis in *Escherichia coli*. *J Bacteriol* 189, 5692–5704 (2007).

Second, the technique that is presented as novel has already published work by Wang et al. (2011), showing opposing results than here in this study, where they observed twisting during growth, a discrepancy mentioned but not discussed in this paper. That being said, I do appreciate the rigor and uses of the technique for more than just showing twisting: it is used to qualitatively and quantitatively describe the dynamics of wall material insertion during growth, These dynamics have been previously explored with other techniques, as mentioned above, which, however, have some limitations compared to the technique presented here. Notably, other techniques, like fluorescent D-amino acids are more perturbative of wall synthesis and their effects on the dynamics of wall properties have not yet been fully elucidated.

Response: We thank Reviewer #2 for recognizing the technical advantage of our study. In Wang et al. 2012 PNAS paper, they demonstrated the use of BFM hook fluorescent labelling but showed the measurement data of only three cells, and under an antibiotic-Cephalexin treatment. The main finding of Wang et al.'s paper was the twisting of cell body during bacterial cell wall growth, which was not observed in our study which investigated a greater number of cells with more systematic and quantitative analyses.

The research results we presented here are not a simple technical repeat of Wang et al.'s method. Instead, by measuring the changes in distance between any pair of these fluorescent 'landmarks' or between a fluorescent BFM and a cell pole using time-lapse microscopy, we offered a new opportunity to investigate the dynamic pattern of murein insertion with high spatial and temporal resolution, while the natural process of bacterial cell wall assembly was not perturbed.

Thus, the only novel finding in this work is how the flagellar motor distribution arises. This is indeed an interesting result, where they find that while the newly produced flagella are non-uniform (center-biased), they become more uniform as they are inherited. In this part of the paper, we can see another advantage of the technique used, which is the high spatial resolution of the labeled flagella in showing wall material inheritance. This allowed the authors to show how flagella that were within spatial resolution limits in one mother cell, were split in two in the daughter cells, an interesting result that can be further explored. While this is a nice result, this is only one panel (and finding) out of four in this paper that is novel. I am confident that, if further fleshed out, studied, and explained, this result itself could stand on its own in a biological journal.

Response: We thank Reviewer #2 for recognizing the merits of our study.

But currently, it is my overall opinion that this work, since it developed a new technique and did

some advanced tracking and modeling on well-studied systems (thus verifying the technique) fits better in a more technique or physics-oriented journal, where such very well done quantitation and modeling, on already described systems, would be better appreciated.

References:

1. Varma, A., Pedro, M. A. de & Young, K. D. FtsZ Directs a Second Mode of Peptidoglycan Synthesis in *Escherichia coli*. *J Bacteriol* 189, 5692–5704 (2007).

2. Pedro, M. A. D., Schwarz, H. & Koch, A. L. Patchiness of murein insertion into the sidewall of *Escherichia coli*. *Microbiology* 149, 1753–1761 (2003).

3. Pedro, M. A. de, Quintela, J. C., Höltje, J. V. & Schwarz, H. Murein segregation in *Escherichia coli*. *J Bacteriol* 179, 2823–2834 (1997).

4. Ursell, T. S. et al. Rod-like bacterial shape is maintained by feedback between cell curvature and cytoskeletal localization. *Proc National Acad Sci* 111, E1025–E1034 (2014).

Response: We thank Reviewer #2 for providing these important references and we are happy to cite these papers in our manuscript.

Based on the above arguments: 1) The research strategy we presented in this work is very innovative; 2) The quantitative aspect of our observations is very novel; 3) We have credited (or will fully credit) the contributions from other researchers, we believe that our work has contributed many new insights into bacterial cell growth and therefore justify publication in *Nature Communications*.

Reviewer #3 (Remarks to the Author):

Regulating the dynamics of bacterial cell growth is highly important as random growth would result in aberrant cell morphology, loss of cell functions or, eventually, inability of division and cell death. However, how and where cells insert new cell-wall material to maintain cell shape and function during growth is still somewhat unclear. One of the problems is that most methods used for identifying the corresponding loci within the cell envelope relies on using labeled D-amino acids, which may not allow appropriate spatiotemporal resolution for the analysis. In this study Sun and coworkers used a clever alternative approach by labeling cell envelope complexes that remain static within the cell envelope. To this end, they used the hooks of the flagellar systems in Escherichia coli for fluorescent labeling, which are distributed around the cell body in a so-called peritrichous flagellation pattern while excluding the cell poles. Once the outer-cell structures, such as the hook are formed, the flagella have been demonstrated to remain static and only shift their general position with respect to the cell and/or other flagella complexes upon cell growth. Thus, the shift of flagellar complexes position can be used to indirectly determine the approximate position of cell-elongation areas and to describe the growth behavior.

Response: We thank Reviewer #3 for the nice summary of our work and for recognizing our technical innovations.

The authors used this set-up to determine, for the first time, the boundary between the inert and active zones for cell wall growth in E. coli, which locates – under the conditions tested – 0.27 μm from the tip of the cell poles. The results further indicated homogenous growth in the axial direction but not the lateral direction so that the diameter of the cell body does not change and not twisting occurs, which fits well to observations made in previous studies. Further growth only occurs at mid-cell upon formation of a new septum upon cell division, so that even flagellar complexes directly localized at mid-cell will not become polar after cell fission. These zones, after cell division, become the new inert zones of the daughter cells. The data obtained were used to formulate a Bernoulli shift map that was employed to predict partitioning of cell wall-anchored complexes (such as the flagella) after cell division in E. coli, which was then challenged experimentally. Newly produced flagella appeared to be center biased, which would allow an even distribution of flagella within one or more generations among the daughter cells.

In my opinion, the authors used a clever approach to determine cell growth dynamics in E. coli, which may now be challenged by studies on other cell species. The authors should mention that, for the time being, this approach is limited to E. coli (in particular in the abstract, line 22). Apart from this, I have no issues with this manuscript.

Response: We thanks Reviewer #3 for the positive and encouraging comments. In our work, we demonstrated the use of cell envelope anchored protein complexes, the bacterial flagellar motors, as fluorescent ‘landmarks’ to study bacterial cell wall growth. We agree with Reviewer #3 that our current conclusions were reached only from experiments on E. coli. We have made this clear in our revised manuscript. (line 23)

REVIEWER COMMENTS

Reviewer #1 (Remarks to the Author):

I commend the author's efforts to promptly address the reviewer questions in midst of COVID-related disruption to work. However, I am not convinced with authors' response to many of the questions. To my surprise, important aspects of the response letter are not incorporated with the revised text, which contains very minimal changes. My specific comments are listed below:

I'd urge the authors to include response fig 1 in the revised manuscript and discuss how the size of the inert zone changes with media nutrient quality (LB vs TB). The authors did not address my query on whether the size of inert zone is related to cell diameter. Since the inert zone is essentially the size of the endocaps, the boundary location should approximately be the length of cell diameter.

While the authors satisfactorily addressed the question on instantaneous velocity vs axial position, the corresponding response fig 2 and the discussions therein should be incorporated in the main text.

I remain unconvinced about the author's claim that there is no relative movement between BFMs during cell elongation. While the data in Response Fig. 5 show that BFMs are anchored to the cell wall, it is not a direct evidence for the absence of relative motion. Could the authors compare the relative elongation measured from the phase contrast image of the cell with the elongation rate inferred from movement of BFMs? I'd also urge the authors to include Response Fig. 5 to support their claim along with the suggested analysis.

I'm still not sure why the authors are plotting Fig 2E, given its a trivial consequence of their definition of H.

Let me clarify that Bernoulli shift map is not a new/novel model, but it follows from the assumption that cell size at division is two times the cell size at birth. We know that such a model is incorrect for *E. coli*, as cell size at division is a constant increment from the cell size at birth (Adder model). Under the adder model, how would the shift map change? I remain unconvinced with the author's claim that Fig 4c is a test of the model as it does not show the predicted cyclic trajectories. Could the authors map out the individual trajectories of the BFMs as the cells undergo divisions?

Reviewer #1 (Remarks to the Author):

I commend the author's efforts to promptly address the reviewer questions in midst of COVID-related disruption to work. However, I am not convinced with authors' response to many of the questions. To my surprise, important aspects of the response letter are not incorporated with the revised text, which contains very minimal changes. My specific comments are listed below:

I'd urge the authors to include response fig 1 in the revised manuscript and discuss how the size of the inert zone changes with media nutrient quality (LB vs TB). The authors did not address my query on whether the size of inert zone is related to cell diameter. Since the inert zone is essentially the size of the endocaps, the boundary location should approximately be the length of cell diameter.

Response: We thank Reviewer #1 for this suggestion. In our revision, we have repeated our experiments to determine the boundary position separating the active and inert zones of cell wall growth when *E. coli* were grown in TB and SOC medium (a rich growth medium). The results, together with the original data collected on *E. coli* grown in LB, were summarized in **Figure 1C and 1D** of the revised manuscript. The sizes of the inert zone were $P_{y0}(\text{LB}) = 0.27 \pm 0.04 \mu\text{m}$, $P_{y0}(\text{TB}) = 0.22 \pm 0.03 \mu\text{m}$, $P_{y0}(\text{SOC}) = 0.37 \pm 0.04 \mu\text{m}$, respectively. Here we see that the size of the inert zone might positively correlate to nutrient quality in the growth media.

We did not find an apparent relationship between the size of the inert zone and cell diameter. First, we determined the cell diameter from the phase contrast images and $D_{\text{LB}} = 1.02 \pm 0.04 \mu\text{m}$, $D_{\text{TB}} = 1.02 \pm 0.06 \mu\text{m}$, $D_{\text{SOC}} = 1.11 \pm 0.06 \mu\text{m}$, where cells grown in SOC medium had a larger diameter. As illustrated in **Response Figure 1A**, the inert zone is approximately the size of the endocaps. However, endocaps are not necessarily or strictly hemispheres (**Response Figure 1B**). Therefore, we conclude the size of the inert zone is roughly proportional to the cell diameter. We have discussed these results in the revised manuscript (Line 74-Line 80).

Response Figure 1. Illustration of *E. coli* cells with different endocap sizes. (A) Endocaps are smaller than half of the cell diameter. (B) Endocaps are hemispheres with a radius equals half of the cell diameter.

While the authors satisfactorily addressed the question on instantaneous velocity vs axial position, the corresponding response fig 2 and the discussions therein should be incorporated in the main text.

Response: We thank Reviewer #1 for this suggestion. In our revision, we have provided the data on instantaneous velocity vs. axial position in **Supplementary Figure S3** and discussed the results in the main text (Line 75).

I remain unconvinced about the author's claim that there is no relative movement between BFMs during cell elongation. While the data in Response Fig. 5 show that BFMs are anchored to the cell wall, it is not a direct evidence for the absence of relative motion. Could the authors compare the relative elongation measured from the phase contrast image of the cell with the elongation rate inferred from movement of BFMs? I'd also urge the authors to include Response Fig. 5 to support their claim along with the suggested analysis.

Response: We thank Reviewer #1 for raising this question and giving us suggestions. We calculated the cell rod growth velocity from the phase contrast image and compared to the growth rate inferred from BFM movements. The data were summarized as Gray points in **Figure 2C**. The cell rod growth data are on the extrapolation of the fitting curves of V_{Dy} vs. D_y (this is identical to measure the relative movement between two BFMs, one at the left boundary and the other one at the right boundary, between the inert and active zones). This again supports that there is no relative movement between BFMs during cell elongation and the cell rod is under uniform growth. We have added a discussion on the newly analyzed data in our revised manuscript (Line106-109).

Following Reviewer #1's suggestion, we have added Response Fig 5 to **Supplementary Figure S2** and discussed in the main text (Line 48-Line 55). We believe these data together support that the BFMs are firmly anchored to cell wall during cell growth and we can probe the dynamics of bacterial cell wall growth by tracing the movement of these fluorescent landmarks.

I'm still not sure why the authors are plotting Fig 2E, given its a trivial consequence of their definition of H.

Response: We thank Reviewer #1 for this suggestion. In our revision, we have moved Fig 2E to **Supplementary Figure S4** for demonstrating all data points.

Let me clarify that Bernoulli shift map is not a new/novel model, but it follows from the assumption that cell size at division is two times the cell size at birth. We know that such a model is incorrect for *E. coli*, as cell size at division is a constant increment from the cell size at birth (Adder model). Under the adder model, how would the shift map change?

Response: As Reviewer #1 point out, Bernoulli shift map is not a new model. We agree with this. However, at the same time, we hope to bring to your attention three important new insights in our Bernoulli shift map model with our experimental validations.

1. For any model to simulate cell growth across generations, the inert cap must be considered and removed from the length calculation.

2. During cell division, the central new endocaps are newly synthesized. The new endocaps are not part of the original rod. Together with point 1, the inert caps must be considered and removed from the length calculation.
3. During cell elongation, expansion of the cylindrical cell rod is homogeneous in the axial direction. Therefore, the position of surface anchored proteins such as BFM's will be fixed in the **normalized coordinate**. After cell division, the BFM's positions will follow Bernoulli shift map in the normalized coordinate, the coordinate referenced by the new cell length. Because we have used the normalized coordinate, it is not necessary that the cells grow up to exactly two times of their original lengths.

For Adder model, cells add a constant size (Δ) between birth and division and have to divide in the middle. In our Bernoulli shift map model, we have used the normalized coordinate and removed the two inert caps, which together overcome the problems of cell length variation. We have added discussion of our Bernoulli shift map model and its incorporation with Adder model in our revised manuscript (Line 243-252).

I remain unconvinced with the author's claim that Fig 4c is a test of the model as it does not show the predicted cyclic trajectories. Could the authors map out the individual trajectories of the BFM's as the cells undergo divisions?

Response: We thank Reviewer #1 for raising this question. We must clarify that **Figure 4B** is the **model prediction** of the moving trajectories of BFM positions across cell generations on the Bernoulli shift map. $N_{y(n)}$ and $N_{y(n+1)}$ are the normalized axial positions of a BFM in the n^{th} and $(n+1)^{\text{th}}$ generations. The dashed line is the guideline for $N_{y(n)} = N_{y(n+1)}$ for continuous mapping. The black lines are the Bernoulli shift map. One dot point on the map represents one cell division with the x axis representing the normalized position of BFM in n^{th} generation and y axis representing the normalized position of BFM in $(n+1)^{\text{th}}$ generation (**Response Figure 2**).

With all due respect, we must clarify that **Figure 4C** is in fact an **experimental validation** of the Bernoulli shift map. In Fig. 4C we recorded $N_{y(n)}$ and $N_{y(n+1)}$ in the normalized coordinates of a **population** of cells ($n = 292$ BFM's) across **only two generations** (one cell division), and we saw the blue dots ($N_{y(n)}, N_{y(n+1)}$) all fell on the Bernoulli shift map (Black lines). One dot represents one BFM normalized positions before and after division.

We agree with Reviewer #1 that a cyclic trajectory will be a best experimental support of the Bernoulli shift map. In **Figure 4D** and **Figure 4E**, we managed to track the BFM's position across more than two generations and the movement of the three representative cases all followed the Bernoulli shift map. Unfortunately, in our experimental setup, it was very hard to track the BFM's position continuously across many generations because cells were overlapping at a high cell density condition, which made an accurate measurement of BFM's position difficult.

Model Prediction

Response Figure 2. Guidance of Fig. 4B, the model prediction of Bernoulli shift map.

REVIEWERS' COMMENTS

Reviewer #1 (Remarks to the Author):

I think the authors have provided a convincing rebuttal and have satisfactorily addressed all my remaining concerns. I think the paper may now be suitable for publication. Congratulations!